# Ionic Liquid-Assisted Fabrication of Bioactive Heterogeneous Magnetic Nanocatalyst with Antioxidant and Antibacterial Activities for the Synthesis of Polyhydroquinoline Derivatives

**DOI:** 10.3390/molecules27051748

**Published:** 2022-03-07

**Authors:** Shefa Mirani Nezhad, Ehsan Nazarzadeh Zare, Azimeh Davarpanah, Seied Ali Pourmousavi, Milad Ashrafizadeh, Alan Prem Kumar

**Affiliations:** 1School of Chemistry, Damghan University, Damghan 36716-41167, Iran; shefamirani@yahoo.com (S.M.N.); a.d.13962017@gmail.com (A.D.); pourmousavi@du.ac.ir (S.A.P.); 2Faculty of Engineering and Natural Sciences, Sabanci University, Orta Mahalle, Üniversite Caddesi No. 27, Orhanlı, Tuzla 34956, Turkey; dvm.milad1994@gmail.com; 3Cancer Science Institute of Singapore and Department of Pharmacology, Yong Loo Lin School of Medicine, National University of Singapore, Singapore 117599, Singapore; apkumar@nus.edu.sg; 4NUS Centre for Cancer Research (N2CR), Yong Loo Lin School of Medicine, National University of Singapore, Singapore 117599, Singapore

**Keywords:** magnetic catalyst, ionic liquid, bioactive, antioxidant, antibacterial, polyhydroquinoline derivatives

## Abstract

Antibacterial materials have obtained much attention in recent years due to the presence of hazardous agents causing oxidative stress and observation of pathogens. However, materials with antioxidant and antibacterial activities can cause toxicity due to their low biocompatibility and safety profile, urging scientists to follow new ways in the synthesis of such materials. Ionic liquids have been employed as a green and environmentally solvent for the fabrication of electrically conductive polymers. In the present study, an antibacterial poly(*p*-phenylenediamine)@Fe_3_O_4_ (P*p*PDA@Fe_3_O_4_) nanocomposite was fabricated using [HPy][HSO_4_] ionic liquid. The chemical preparation of P*p*PDA@Fe_3_O_4_ nanocomposite was initiated through the oxidative polymerization of *p*-phenylenediamine by ammonium persulfate in the presence of [HPy][HSO_4_]. The P*p*PDA@Fe_3_O_4_ nanocomposite exhibited antibacterial properties against Gram-negative (*Escherichia coli*) and Gram-positive (*Bacillus subtilis*) bacteria. The P*p*PDA@Fe_3_O_4_ nanocomposite was employed as a heterogeneous nanocatalysis for one-pot synthesis of polyhydroquinoline derivatives using aromatic aldehyde, dimedone, benzyl acetoacetate, and ammonium acetate. Polyhydroquinoline derivatives were synthesized in significant yields (90–97%) without a difficult work-up procedure in short reaction times. Additionally, P*p*PDA@Fe_3_O_4_ nanocatalyst was recycled for at least five consecutive catalytic runs with a minor decrease in the catalytic activity. In this case, 11 derivatives of polyhydroquinoline showed in vitro antioxidant activity between 70–98%.

## 1. Introduction

The ionic liquids (ILs) are considered substances with a melting point below 100 °C composed of a cation and an anion [1]. The ions present in ILs are tunable and it is possible to develop solvents and catalysts from ILs. The early reports of using ILs in enzymatic reactions were performed in 2000 [2,3]. In addition to chemical synthesis and catalysis, ILs have been employed in electrochemistry, biotechnology, and pharmaceutics [4]. The ILs have been utilized as stabilizers for DNA storage [5]. ILs have been employed in chemical investigations owing to their nonvolatility, high thermal and electrochemical stability [6]. They are environmentally friendly salts that have been employed in the synthesis of organic compounds, catalytic reactions as well as the synthesis of intrinsically conductive polymers (ICPs) [7,8]. 

Intrinsically conductive polymers (ICPs) such as polyaniline, polypyrrole, and polythiophene, owning to their electrical conductivity, biocompatibility, and environmental stability, have been widely applied in different arenas, e.g., water treatment, catalysts, and biomedical applications [9,10]. Among ICPs, the polyaniline derivatives such as poly(phenylenediamines) (ortho, meta, and para) have attracted special attention. Poly(phenylenediamines) are highly aromatic ladder polymers that exhibited good solubility and poor electrical conductivity compared to polyaniline [11,12,13]. They have been widely used in water treatment, biosensors, and biomedical applications [14]. In order to improve electrical conductivity, thermal stability, and viscosity of poly(phenylenediamines), ILs can be utilized [15]. Recently, ILs such as 1-alkyl-3-methylimidazolium, 1-butyl-4-methylpyridinium, quaternary ethyltributylammonium, pyridinium carboxylic acid sulfate have been employed for the synthesis of polyaniline and its derivatives [8,11,16,17]. It was reported that ILs played the role of lubricants, plasticizers, interfacial agents in polymer systems, generating enhancements in the mechanical properties, solubility, electrical conductivity, crystallinity, and thermal stability [15].

Iron oxide is an FDA-approved agent significantly applied in nanomedicine [5]. The iron oxide nanomaterials have a diameter of 15–100 nm composed of magnetite or maghemite and have shown great biomedical applications as contrast materials, drug carriers, and thermal-based therapeutics [18]. Iron oxide nanoparticles are under attention because of their excellent properties in polymer-based catalysts [19,20]. This is because iron oxide nanoparticles possess a large surface area for substrate molecules. Furthermore, after the completion of the reactions, the magnetic catalysts can be separated easily from the solution using an external magnet. Additionally, magnetic catalysts can be reused up to numerous runs almost without loss of catalytic activity [21,22].

Several papers have reported the use of P*p*PDA composites as an effective catalyst in organic reactions. For example, Cu_2_O-Cu(OH)_2_/P*p*PDA nanocomposite was used as a high-efficiency catalyst for methanol electrooxidation [23]. P*p*PDA/carbon black composite was employed for oxygen reduction [24]. ZnCr-layered double hydroxides/P*p*PDA/Cu(II) as a catalyst for the synthesis of pyrrole derivatives [25]. In another work, layered double hydroxides/P*p*PDA was applied as an effective catalyst for the synthesis of indolizines [26].

Hantzsch products (e.g., 1,4-dihydropyridine, polyhydroquinoline, and acridine) are products with significant biological activities. They have pharmacological properties, e.g., vasodilator, antihypertensive, bronchodilator, anti-atherosclerotic, hepatoprotective, anti-tumor, anti-mutagenic, neuroprotective, and anti-diabetic [27]. 4-Substituted 1,4-dihydropyridines (1,4-DHPs) establishes a significant class of Ca^2+^ channel blockers [28] and has been used as one of the most important classes of drugs for cardiovascular disease treatment [29]. Polyhydroquinolines have been prepared through conventional heating, microwave irradiation [30], and ultrasound [31].

Herein, we designed a unique organic-inorganic antibacterial and antioxidant nanocatalyst based on P*p*PDA and iron oxide nanoparticles with assisted [HPy][HSO_4_] using ammonium persulfate as an oxidant. The prepared P*p*PDA@Fe_3_O_4_ bioactive nanocomposite was employed as an efficient retrievable eco-friendly catalyst for the synthesis of polyhydroquinoline derivatives through the one-pot four-component reaction of dimedone, benzyl acetoacetate, ammonium acetate, and different aromatic aldehydes under mild reaction conditions (Figure 1).

## 2. Results

### 2.1. Characterization of Polymer-Based Catalyst

**FTIR***:* The FTIR spectra of the prepared P*p*PDA and P*p*PDA@Fe_3_O_4_ nanocomposite in the presence of the [HPy][HSO_4_] ionic liquid are shown in Figure 1a. The P*p*PDA and P*p*PDA@Fe_3_O_4_ nanocomposite showed very similar spectra with tiny differences. In the FTIR spectrum of P*p*PDA, the absorption peak at 3200–3450 cm^−1^ is related to the stretching vibration of the NH_2_ and N-H groups. Two characteristic absorption peaks at 1570 cm^−1^ and 1503 cm^−1^ are associated with the stretching mode of quinoid imine and benzenoid amine units, respectively [32]. The absorption peaks with different intensities in the areas of 1310 cm^−1^, 1108–1118 cm^−1^, and 830 cm^−1^ are related to the stretching vibrations of SO_4_, S=O, and S-OH in the [HPy][HSO_4_] ionic liquid, respectively [11]. The incorporation of iron oxide nanoparticles in the P*p*PDA matrix led to the appearance of an obvious absorption peak at around 505 cm^−1^ that related to the Fe-O-Fe stretching modes in Fe_3_O_4_ [33]. 

**Elemental analysis***:* Elemental analysis (CHNSO) was employed for characterization of the prepared P*p*PDA and P*p*PDA@Fe_3_O_4_ nanocomposite in the presence of the [HPy][HSO_4_] ionic liquid (Table 1). According to the data in Table 1, the existence of sulfur and oxygen elements in the P*p*PDA approved the presence of [HPy][HSO_4_] ionic liquid in the structure of the P*p*PDA. The increase of oxygen element in the P*p*PDA@Fe_3_O_4_ nanocomposite compared with P*p*PDA shows the presence of Fe_3_O_4_ nanoparticles in the nanocomposite [11].

**EDX**: The chemical composition of the prepared P*p*PDA and P*p*PDA@Fe_3_O_4_ nanocomposite in the presence of the [HPy][HSO_4_] ionic liquid was also estimated using the EDX technique as revealed in Figure 1b. The comparison of spectra and tabulated data indicated the existence of different amounts of O, C, N, and S elements. The presence of S in both P*p*PDA and P*p*PDA@Fe_3_O_4_ nanocomposite is related to [HPy][HSO_4_] ionic liquid. In addition, the existence of Fe in the P*p*PDA@Fe_3_O_4_ sample indicates the iron oxide in the composite [34]. 

**X-ray diffraction**: XRD patterns of the prepared P*p*PDA and P*p*PDA@Fe_3_O_4_ nanocomposite in the presence of the [HPy][HSO_4_] ionic liquid are shown in Figure 1c. According to the literature, the XRD pattern of Fe_3_O_4_ nanoparticles indicated a crystalline nature [35]. A semicrystalline nature was observed in the XRD pattern of P*p*PDA owing to intermolecular interactions between the P*p*PDA chains and the [HPy][HSO_4_] ionic liquid [36]. The XRD pattern of P*p*PDA@Fe_3_O_4_ nanocomposite showed more crystallinity compared with P*p*PDA due to the presence of crystalline nanoparticles of Fe_3_O_4_ [37].

**FESEM***:* The morphology of Fe_3_O_4_ nanoparticles, P*p*PDA, and P*p*PDA@Fe_3_O_4_ nanocomposite was investigated using FESEM. The FESEM micrographs of the prepared Fe_3_O_4_ nanoparticles (**a**), P*p*PDA (**b**), and P*p*PDA@Fe_3_O_4_ nanocomposite (**c**) are shown in Figure 2. The FESEM micrograph of Fe_3_O_4_ nanoparticles shows a spherelike structure with a diameter of ~50 nm. Polyhedral shapes with diameters of ~150 nm and ~100 nm observe in the FESEM micrographs of P*p*PDA, and P*p*PDA@Fe_3_O_4_ nanocomposite, respectively.

**VSM***:* Vibrating sample magnetometer (VSM) was employed for the evaluation of the magnetic property of P*p*PDA@Fe_3_O_4_ nanocomposite as shown in Figure 2d. In the VSM of the P*p*PDA@Fe_3_O_4_ nanocomposite, the amount of magnetic coercivity (Hc) and magnetic remanence (Mr) is equal to zero. This indicates the P*p*PDA@Fe_3_O_4_ nanocomposite has a superparamagnetic property with a magnetization saturation value of 30.01 emu/g [38].

**Solubility test*:*** The solubility of the prepared P*p*PDA in the presence and absence of [HPy][HSO_4_], as well as the P*p*PDA@Fe_3_O_4_ magnetic nanocomposite prepared in [HPy][HSO_4_] in different solvents, were studied (Table 2). The results indicate that the presence of [HPy][HSO_4_] improves the solubility of the P*p*PDA. On the other hand, the presence of Fe_3_O_4_ nanoparticles in P*p*PDA leads to reducing in the solubility of the P*p*PDA.

### 2.2. Evaluation of the Catalytic Activity of PpPDA@Fe_3_O_4_ Nanocomposite

In the current study, we offered a new and efficient technique for the synthesis of polyhydroquinolines using P*p*PDA@Fe_3_O_4_ nanocomposite. We investigated the four-component Hantzsch condensation by aromatic aldehyde, dimedone, benzyl acetoacetate, and ammonium acetate.

The effect of various solvents on the reaction rate and yield of the products was investigated to optimize the reaction conditions.

To optimize the reaction conditions, firstly the various solvents’ effect on the rate of reaction and products yield was investigated. The reaction of benzaldehyde (**1**, 1 mmol), dimedon, (**2**, 1 mmol), ammonium acetate (**3**, 1 mmol), and benzyl acetoacetate (**4**, 1 mmol) as a model reaction was catalyzed by 0.03 g of P*p*PDA@Fe_3_O_4_ in different solvents, e.g., water, ethanol (EtOH), chloroform (CHCl_3_), tetrahydrofuran (THF) and hexane at reflux conditions (Table 3). In aprotic solvents, e.g., CHCl_3_, THF, and hexane, the reaction rate was very slow and product yield was low whereas reaction rates, as well as product yields in protic solvents, were improved. In water and solvent-free conditions, the expected product was achieved only in low yield after 4 h. Furthermore, the above condensation reaction by the P*p*PDA@Fe_3_O_4_ catalyst was also carried out in ethanol at reflux conditions. 

To optimize the temperature of the reaction, the mixture was heated at various temperatures. The yield of the products was increased when the reaction temperature was raised from room temperature to reflux conditions. Moreover, when 0.02, 0.03, 0.04, and 0.06 of P*p*PDA@Fe_3_O_4_ nanocatalyst were used, the yield of the products was 60%, 85%, 94%, and 96%, respectively. Therefore, 0.04 g of P*p*PDA@Fe_3_O_4_ was an optimal amount for producing products with high yields. For better comparison, the reaction was also investigated in the absence of the catalyst. Results (Table 3) showed that the rate of reaction was very slow and product yield was low.

For better comparison, the synthesis of polyhydroquinoline derivatives was studied in the absence and presence of various catalysts, e.g., P*p*PDA, P*p*PDA@[HPy][HSO_4_], P*p*PDA@Fe_3_O_4_@[HPy][HSO_4_] under the same conditions, and the results are shown in Table 4. The results display that the P*p*PDA@[HPy][HSO_4_] has better catalytic properties than the P*p*PDA. The major problem of P*p*PDA catalyst is easy solubility in organic solvents and therefore its isolation is difficult. In addition, results showed that the use of 60 mg of P*p*PDA@Fe_3_O_4_@[HPy][HSO_4_] catalyst under the same reaction conditions reduce the reaction time and increased the reaction yield. Consequently, the P*p*PDA@Fe_3_O_4_@[HPy][HSO_4_] nanocatalyst was selected as an effective catalyst to perform the reactions.

Table 5 shows that aromatic aldehydes containing electron-donating and electron-withdrawing groups reacted with dimedone, benzyl acetoacetate, and ammonium acetate in the presence of P*p*PDA@Fe_3_O_4_ magnetic nanocatalyst in optimal conditions and in a short time to produced polyhydroquinolines with excellent yields. Likewise, thiophene-carbaldehyde and furfural (heteroaromatic aldehydes) produced the desired product after 35 min with 92% and 90% yields respectively (Table 5, entries 12 and 14). P*p*PDA@Fe_3_O_4_ magnetic nanocatalyst was also suitable for the synthesis of polyhydroquinolines from aliphatic aldehyde such as α-methyl cinnamaldehyde (Table 5, entry 13). Dialdehydes such as para phenylene dialdehyde and 2,2′-(hexane-1,6-diylbis(oxy))dibenzaldehyde reacted under optimal conditions using P*p*PDA@Fe_3_O_4_ magnetic nanocatalyst and produced products with high yields (Table 5, entries 23 and 24). 

**Proposed mechanistic scheme*:*** The P*p*PDA@Fe_3_O_4_ nanocomposite with active sites such as base (secondary amines in polymer backbone), Brønsted acid ([HPy][HSO_4_], Lewis acid site (Fe^3+^ in Fe_3_O_4_), and large surface area, play a significant role in all steps of reactions as demonstrated in Figure 3. First, dimedone is activated in the presence of amine groups in P*p*PDA, and then, as a nucleophile, attacks the aldehyde activated by the [HPy][HSO_4_] to form intermediate (I) (Knoevenagel condensation). On the other hand, ammonium acetate is converted to acetic acid and ammonia, and then the ammonia as a nucleophile attacks the benzyl acetate activated by the [HPy][HSO_4_] to form intermediate (II). In the next step, Michael’s addition reaction of intermediate (II) to intermediate (I) causes the formation of intermediate (III). The intermediate (III) is then converted to the intermediate (IV) by tautomerization and the product (VI) is obtained after cyclization reaction.

**Recovery and reusability**: Recyclability is an important property of heterogeneous catalytic systems in terms of environmental protection and industrial application. To evaluate the reusability of P*p*PDA@Fe_3_O_4_, it was magnetically isolated from the reaction mixture, washed several times with distilled water and ethanol, dried at room temperature, utilized again in the next reaction. As is observed in Figure 4, the yield of the products was not reduced considerably after five successive catalytic runs and the catalyst has retained its efficacy and stability in the synthesis of polyhydroquinolines derivatives.

**Antioxidant activity***:* The antioxidant activity of Fe_3_O_4_ NPs, P*p*PDA, P*p*PDA@[HPy][HSO_4_], P*p*PDA@Fe_3_O_4_@[HPy][HSO_4_], and synthesized polyhydroquinoline derivatives was studied using the 2,2-diphenyl-1-picrylhydrazyl (DPPH)radical scavenging model (Figure 5). Results showed that all used materials to prepare magnetic nanocatalysts had antioxidant activities between 72% and 90%. In addition, the antioxidant activity of 24 synthesized derivatives was investigated. Only 11 derivatives showed antioxidant activity between 75% and 98%. These results suggest that these derivatives may play a role in the synthesis of immune-boosting drugs. 

**Antibacterial activity:** The *in vitro* antibacterial activities of the Fe_3_O_4_ NPs, P*p*PDA, P*p*PDA@[HPy][HSO_4_] (P*p*PDA@IL), P*p*PDA@Fe_3_O_4_@[HPy][HSO_4_] (nanocatalyst), and seven of polyhydroquinoline derivatives (5c, 5i, 5j, 5r, 5s, 5o, and 5v) were investigated against *Escherichia coli* and *Bacillus subtilis*, and results shown in Table 6 and Figure 6. The results showed that the bare P*p*PDA and Fe_3_O_4_ NPs had good growth inhibitory effects against tested microorganisms, and among them, Fe_3_O_4_ NPs exhibited the highest antibacterial activity against both microorganisms while P*p*PDA@IL was effective against *Bacillus subtilis*. The nanocatalyst showed lower antimicrobial activity than that of bare minerals against tested bacteria. Moreover, among polyhydroquinoline derivatives, only 5r and 5v had good growth inhibitory effects against tested microorganisms while 5i and 5o showed no effect against tested microorganisms.

**Spectroscopic Data** (^1^H-NMR and ^13^C-NMR of products are shown in Appendix A)

*Benzyl2,7,7-trimethyl-4-(4-nitrophenyl)-5-oxo-1,4,5,6,7,8-hexahydroquinoline-3-carboxylate* (Table 5, Entry 2, Appendix A)

Solid, m.p.148–150 °C. ^1^H NMR (400 MHz, DMSO-*d*_6_): *δ* (ppm) 0.80 (s, 3H, CH_3_), 1.00 (s, 3H, CH_3_), 1.97 (d, 1H, *J* = 16 Hz, CH_2_), 2.18 (d, 1H, *J* = 16 Hz, CH_2_), 2.32 (s, 3H, CH_3_), 2.27–2.246 (m, 2H, CH_2_), 4.96–5.61 (m, 1H, benzilic, OCH_2_ benzilic), 7.16–7.37 (m, 7H, aromatic), 8.04 (d, 2H, *J* = 8 Hz), 9.33 (s, 1H, NH), ^13^C-NMR (100 MHz, DMSO-*d*_6_): *δ* (ppm) 18.93, 26.87, 29.46, 32.62, 37.06, 40.58, 45.41, 50.50, 65.38, 102.34, 109.59, 123.63, 128.29, 128.73, 129.25, 136.93, 146.07, 147.44, 150.47, 155.27, 166.61, 194.77.

*Benzyl4-(3-ethoxy-4-hydroxyphenyl)-2,7,7-trimethyl-5-oxo-1,4,5,6,7,8-hexahydroquinoline-3-carboxylate* (Table 5, Entry 3, Appendix A)

Solid, m.p.209–211 °C. ^1^H NMR (400 MHz, DMSO-*d*_6_): *δ* (ppm) 0.85(s, 3H, CH_3_), 1.00 (s, 3H, CH_3_), 1.24 (t, 3H, *J* = 6.8 Hz CH_3_), 1.97 (d, 1H, *J* = 16.4 Hz, CH_2_), 2.13 (d, 1H, *J* = 16.4 Hz, CH_2_), 2.29 (s, 3H, CH_3_), 2.24–2.29 (m, 2H, CH_2_), 2.41(d, 2H, *J* = 17.2 Hz, CH_2_), 4.77 (s, 1H, benzilic), 5.33 (AB q, 2H, *J* = 12 Hz, OCH_2_ benzilic), 6.46–6.63 (m, 3H, aromatic), 7.22–7.33 (m, 5H, aromatic), 8.56 (s, 1H, OH) 9.07 (s, 1H, NH), ^13^C-NMR (100 MHz, DMSO-*d*_6_): *δ* (ppm) 15.23, 18.58, 26.82, 29.67, 32.60, 35.41, 50.76, 64.06, 65.22, 103.90, 110.92, 113.96, 115.49, 120.06, 128.11, 128.16, 128.75, 137.19, 139.30, 145.29, 145.84, 146.23, 149.52, 167.28, 194.88. 

*Benzyl2,7,7-trimethyl-5-oxo-4-(3,4,5-trimethoxyphenyl)-1,4,5,6,7,8-hexahydroquinoline-3-carboxylate* (Table 5, Entry 5, Appendix A)

Solid, m.p.140–142 °C. ^1^H NMR (400 MHz, DMSO-*d*_6_): *δ* (ppm) 0.91 (s, 3H, CH_3_), 1.02 (s, 3H, CH_3_), 2.02 (d, 1H, *J* = 16 Hz, CH_2_), 2.19 (d, 1H, *J* = 15.6 Hz, CH_2_), 2.31 (s, 3H, CH_3_), 2.31–2.47 (m, 2H, CH_2_), 3.37 (s, 6H, OCH_3_), 3.55 (s, 3H, OCH_3_), 4.86 (s, 1H, benzilic), 5.08 (AB q, 2H, *J* = 12 Hz, OCH_2_ benzilic), 6.37 (s, 2H, aromatic), 7.25–7.33 (m, 5H, aromatic), 9.17 (s, 1H, NH). ^13^C-NMR (100 MHz, DMSO-*d*_6_): *δ* (ppm) 18.90, 19.03, 21.51, 26.75, 29.70, 31.62, 32.61, 36.20, 50.70, 55.98, 60.33, 65.29, 66.07, 103.57, 105.05, 110.37, 126.87, 127.61, 127.85, 128.22, 128.79, 136.13, 137.21, 143.75, 146.30, 150.13, 167.19, 172.53, 194.96.

*Benzyl2,7,7-trimethyl-5-oxo-4-(p-tolyl)-1,4,5,6,7,8-hexahydroquinoline-3-carboxylate* (Table 5, Entry 9, Appendix A)

Solid, m.p.165–167 °C. ^1^H-NMR (400 MHz, DMSO-*d*_6_): *δ* (ppm) 0.83 (s, 3H, CH_3_), 1.00 (s, 3H, CH_3_), 1.96 (d, 1H, *J* = 16 Hz, CH_2_), 2.13 (d, 1H, *J* = 16 Hz, CH_2_), 2.20 (s, 3H, CH_3_), 2.25 (s, 3H, CH_3_), 2.25–2.39 (m, 2H, CH_2_), 2.41(d, 2H, *J* = 30 Hz, CH_2_), 4.84 (s, 1H, benzilic), 5.01 (AB q, 2H, *J* = 12.8 Hz, OCH_2_ benzilic), 6.96 (d, 2H, *J* = 8 Hz, aromatic), 7.01 (d, 2H, *J* = 8 Hz, aromatic), 7.19–7.32 (m, 5H, aromatic), 9.12 (s, 1H, NH), ^13^C-NMR (100 MHz, DMSO-*d*_6_): *δ* (ppm) 18.87, 21.07, 21.78, 26.90, 29.61, 32.60, 35.20, 39.32, 46.93, 50.70, 65.22, 103.68, 110.72, 128.18, 128,74, 135.07, 137.14, 145.11, 146.12, 149.70, 167.12, 172.73, 194.77. 

*Benzyl2,7,7-trimethyl-5-oxo-4-(p-tolyl)-1,4,5,6,7,8-hexahydroquinoline-3-carboxylate* (Table 5, Entry 10, Appendix A) 

Solid, m.p.126–128 °C. ^1^H NMR (400 MHz, DMSO-*d*_6_): *δ* (ppm) 0.83 (s, 3H, CH_3_), 1.00 (s, 3H, CH_3_), 1.97 (d, 1H, *J* = 15.6 Hz, CH_2_), 3.94 (d, 1H, *J* = 16.4 Hz, CH_2_), 2.30 (s, 3H, CH_3_), 2.41 (s, 3H, SCH_3_), 2.25–2.42 (m, 2H, CH_2_), 2.53 (d, 2H, *J* = 20 Hz, CH_2_), 4.83 (s, 1H, benzilic), 5.02 (AB q, 2H, *J* = 12 Hz, OCH_2_ benzilic), 7.06–7.30 (m, 9H, aromatic), 9.15 (s, 1H, NH), ^13^C-NMR (100 MHz, DMSO-*d*_6_): *δ* (ppm) 15.31, 18.88, 26.95, 29.57, 32.62, 35.85, 50.67, 65.23, 103.41, 110.47, 125.62, 126.08, 128.18, 128.64, 128.75, 135.30, 137.12, 144.90, 146.320, 146.80, 167.02, 194.79.

*Benzyl4-(4-ethylphenyl)-2,7,7-trimethyl-5-oxo-1,4,5,6,7,8-hexahydroquinoline-3-carboxylate* (Table 5, Entry 13, Appendix A) 

Solid, m.p.165–167 °C. ^1^H NMR (400 MHz, DMSO-*d*_6_): *δ* (ppm) 0.84 (s, 3H, CH_3_), 1.00 (s, 3H, CH_3_), 1.13 (t, 3H, *J* = 8 Hz, CH_3_), 1.97 (d, 1H, *J* = 15.6 Hz, CH_2_), 2.15 (d, 1H, *J* = 16.4 Hz, CH_2_), 2.29 (s, 3H, CH_3_), 2.26–2.48 (m, 2H, CH_2_), 4.84 (s, 1H, benzilic), 5.02 (AB q, 2H, *J* = 12 Hz, OCH_2_ benzilic), 7.01 (dd, 4H, *J* = 8, 15.6 aromatic),7.18–7.41 (m, 4H, aromatic) 9.11 (s, 1H, NH), ^13^C-NMR (100 MHz, DMSO-*d*_6_): *δ* (ppm) 16.09, 18.90, 27.01, 28.23, 29.57, 32.62, 35.81, 50.72, 65.21, 103.74, 110.71, 127.60, 127.94, 128.16, 128.72, 137.17, 141.43, 145.38, 146.11, 149.75, 167.13, 194.77.

*Benzyl4-(3,4-dimethoxyphenyl)-2,7,7-trimethyl-5-oxo-1,4,5,6,7,8-hexahydroquinoline-3-carboxylate* (Table 5, Entry15, Appendix A) 

Solid, m.p.151–153–167 °C. ^1^H NMR (400 MHz, DMSO-*d*_6_): *δ* (ppm) 0.86 (s, 3H, CH_3_), 1.01 (s, 3H, CH_3_), 1.99 (d, 1H, *J* = 16 Hz, CH_2_), 2.17 (d, 1H, *J* = 16.4 Hz, CH_2_), 2.31 (s, 3H, CH_3_), 2.19–2.50 (m, 2H, CH_2_), 247 (d, 2H, *J* = 24 Hz, CH_2_), 3.53 (s, 3H, OCH_3_), 3.67 (s, 3H, OCH_3_), 4.84 (s, 1H, benzilic), 5.04 (AB q, 2H, *J* = 12 Hz, OCH_2_ benzilic), 6.61 (dd, 1H, *J* = 2 Hz, 8 Hz, H aromatic), 6.70–6.76 (m, 2H, aromatic), 7.22–7.24 (m, 2H, aromatic), 7.29–7.32 (m, 3H, aromatic) 9.14 (s, 1H, NH).

*Benzyl4-(2-chlorophenyl)-2,7,7-trimethyl-5-oxo-1,4,5,6,7,8-hexahydroquinoline-3-carboxylate* (Table 5, Entry 18, Appendix A)

Solid, m.p.178–180 °C. ^1^H NMR (400 MHz, DMSO-*d*_6_): *δ* (ppm) 0.84(s, 3H, CH_3_), 1.00 (s, 3H, CH_3_), 1.91 (d, 1H, *J* = 16 Hz, CH_2_), 2.14 (d, 1H, *J* = 16 Hz, CH_2_), 2.28 (s, 3H, CH_3_), 2.25–2.240 (m, 2H, CH_2_), 4.99 (AB q, 2H, *J* = 12 Hz, OCH_2_ benzilic), 5.21 (s, 1H, benzilic), 7.08–7.27 (m, 9H, aromatic), 9.17 (s, 1H, NH), ^13^C-NMR (100 MHz, DMSO-*d*_6_): *δ* (ppm) 18.86, 21.70, 26.82, 29.61, 32.44, 35.49, 46.87, 50.69, 56.52, 56.03, 101.83, 103.14, 110.13, 127.72,128.09, 128.65, 129.56, 132.07, 132.56, 137.27, 145.47, 146.37, 150.12, 166.99, 172.64, 194.41.

*Benzyl4-(3-hydroxyphenyl)-2,7,7-trimethyl-5-oxo-1,4,5,6,7,8-hexahydroquinoline-3-carboxylate* (Table 5, Entry 19, Appendix A) 

Solid, m.p.212–214 °C. ^1^H NMR (400 MHz, DMSO-*d*_6_): *δ* (ppm) 0.85 (s, 3H, CH_3_), 1.00 (s, 3H, CH_3_), 1.97 (d, 1H, *J* = 16 Hz, CH_2_), 2.13 (d, 1H, *J* = 16 Hz, CH_2_), 2.30 (s, 3H, CH_3_), 2.24–2.230 (m, 2H, CH_2_), 4.89 (s, 1H, benzilic), 5.04 (AB q, 2H, *J* = 12Hz, OCH_2_ benzilic), 6.48 (dd, 1H, *J* = 1.6 Hz, 8 Hz, aromatic), 7.56–6.61 (m, 2H, aromatic), 6.94 (t, 1H, *J* = 8 Hz, aromatic), 7.18–7.21 (m, 2H, aromatic), 7.28–7.31 (m, 4H, aromatic, OH), 9.11 (s, 1H, NH), ^13^C-NMR (100 MHz, DMSO-*d_6_*): *δ* (ppm) 18.91, 26.99, 29.57, 32.61, 35.96, 50.75, 56.51, 65.17, 103.45, 110.60, 113.16, 113.16, 115.02, 118.73, 128.01, 128.10, 128.76, 1129.03, 137.21, 146.18, 149.25, 149.74, 157.42, 167.17, 194.79. 

*Benzyl4-(2-hydroxy-3-methoxyphenyl)-2,7,7-trimethyl-5-oxo-1,4,5,6,7,8-hexahydroquinoline-3-carboxylate* (Table 5, Entry 20, Appendix A) 

Solid, m.p.248–250 °C. ^1^H NMR (400 MHz, DMSO-*d_6_*): *δ* (ppm) 0.78 (s, 3H, CH_3_), 1.01 (s, 3H, CH_3_), 2.09 (d, 1H, *J* = 16 Hz, CH_2_), 2.15 (d, 1H, *J* = 16 Hz, CH_2_), 2.41–2.45 (m, 5H, CH_2,_ CH_3_), 3.70 (s, 3H, OCH_3_), 4.97 (AB q, 2H, *J* = 13.2 Hz, OCH_2_ benzilic), 5.08 (s, 1H, benzilic), 6.54 (dd, 1H, *J* = 1.6 Hz, 7.6 Hz, H aromatic), 6.65–6.73 (m, 2H, aromatic), 6.99–7.01 (m, 2H, aromatic), 7.19–7.23 (m, 3H, aromatic), 9.26 (s, 1H, OH), 9.43 (s, 1H, NH).

*Benzyl4-(3-hydroxy-4-methoxyphenyl)-2,7,7-trimethyl-5-oxo-1,4,5,6,7,8-hexahydroquinoline-3-carboxylate* (Table 5, Entry 22, Appendix A)

Solid, m.p.141–143 °C. ^1^H NMR (400 MHz, DMSO-*d*_6_): *δ* (ppm) 0.85 (s, 3H, CH_3_), 1.00 (s, 3H, CH_3_), 1.98 (d, 1H, *J* = 16 Hz, CH_2_), 2.24–2.41 (m, 6H, CH_2,_ CH_3_), 3.68 (s, 3H, OCH_3_), 4.78 (s, 1H, benzilic), 5.03 (AB q, 2H, *J* = 13.6 Hz, OCH_2_ benzilic), 6.51(dd, 1H, *J* = 2 Hz, 8.4 Hz, H aromatic), 6.62–6.70 (m, 2H, aromatic), 7.19–7.21 (m, 2H, aromatic), 7.28–7.31 (m, 3H, aromatic), 8.68 (s, 1H, OH), 9.08 (s, 1H, NH), ^13^C-NMR (100 MHz, DMSO-*d*_6_): *δ* (ppm) 18.89, 27.03, 29.58, 32.60, 35.31, 50.78, 56.02, 65.13, 103.78, 110.83, 112.05, 115.61, 118.54, 128.01, 128.09, 128.75, 137.24, 140.86, 145.81, 146.18, 146.30, 149.45, 167.25, 194.83.

*Dibenzyl4,4′-(1,4-phenylene)bis(2,7,7-trimethyl-5-oxo-1,4,5,6,7,8-hexahydroquinoline-3-carboxylate)* (Table 5, Entry 23, Appendix A)

Solid, m.p.327–330 °C. ^1^H NMR (400 MHz, DMSO-*d_6_*): *δ* (ppm) 0.81(s, 3H, CH_3_), 0.98 (s, 3H, CH_3_), 1.98 (d, 1H, *J* = 16 Hz, CH_2_), 2.10–2.15 (m, 1H, *J* = 16 Hz, CH_2_), 2.30 (s, 3H, CH_3_), 2.29–2.236 (m, 2H, CH_2_), 4.84 (s, 1H, benzilic), 4.99 (AB q, 2H, *J* = 12 Hz, OCH_2_ benzilic), 6.89–6.90 (m, 2H, aromatic), 7.12–7.17 (m, 2H, aromatic), 7.26–7.28 (m, 3H, aromatic), 9.12 (s, 1H, NH), ^13^C-NMR (100 MHz, DMSO-*d_6_*): *δ* (ppm) 26.84, 27.36, 29.30, 29.57, 32.64, 35.26, 35.47, 45.81, 50.70, 65.20, 103.35, 103.49, 110.51, 110.68, 127.95, 128.12, 128.71, 137.10, 145.10, 145.30, 146.37, 149.79, 150.00, 167.16, 194.88. 

*Dibenzyl4,4′-((hexane-1,6-diylbis(oxy))bis(2,1-phenylene))bis(2,7,7-trimethyl-5-oxo-1,4,5,6,7,8-hexahydroquinoline-3-carboxylate)* (Table 5, Entry24, Appendix A)

Solid, m.p.161–163 °C. ^1^H NMR (300 MHz, DMSO-d6): δ = 0.84 (s, 3H, CH_3_), 1.00 (s, 3H, CH_3_), 1.49 (br, 2H, CH_2_ bridge), 1.72 (br, 2H, CH_2_ bridge), 1.90 (d,1H, J = 16 Hz, CH_2_), 2.13–2.39 (m, 5H, J = 18 Hz, CH_2_, CH_3_), 2.48 (d, 1H, J = 16 Hz CH_2_), 3.82 (dd, 2H, J = 7.2, 38.4 Hz OCH_2_ bridge), 4.97 (AB q, 2H, J = 12 Hz, OCH_2_), 5.08 (s, 1H, benzlic), 6.68–6.84 (m, 2H, aromatic), 7.03–7.29 (m, 7H, aromatic), 9.02 (s, 1H, NH), ^13^C NMR(300 MHz, DMSO-d_6_): δ = 18.92, 26.13, 26.63, 29.71, 29.83, 32.40, 34.77, 50.87, 64.99, 67.93, 102.59, 109.29, 112.10, 119.57, 127.44, 128.04, 128.13, 128.66, 132.09, 134.71, 137.33, 145.50, 150.05, 157.57, 167.53, 194.37 ppm.

## 3. Experimental Section

### 3.1. Materials

Paraphenylenediamine (*p*PDA), ammonium persulfate (APS), pyridine, sulfuric acid, dimedone, Iron (II), and (III) slats, benzyl acetoacetate, ammonium acetate, DPPH (2,2-diphenyl-1-picrylhydrazyl) radical scavenging, and all employed solvents were provided by Merck company (Germany).

### 3.2. Preparation of Iron Oxide Nanoparticles

Iron oxide nanoparticles (Fe_3_O_4_ NPs) were prepared by the co-precipitation technique as follows [40]. FeCl_3_·6H_2_O (2.73 g) and FeCl_2_·4H_2_O (0.99 g) were dissolved in deionized water at ambient temperature, and then 10 mL ammonia solution (25%) was added into the above solution under constant stirring for a half-hour, and the final pH was 10. Lastly, the black precipitate was isolated by a magnet and washed with distilled water and ethanol, and dried at 80 ^◦^C under vacuum for 2 h.

### 3.3. Pyridinium Hydrogen Sulfate [HPy][HSO_4_] Preparation

Pyridinium hydrogen sulfate [HPy][HSO_4_] was prepared as follows (Figure 2a) [41]. 10 mL of pyridine was poured into a flask, then 6.76 mL of sulfuric acid solution was added slowly into pyridine for one hour under stirring at 0–5 °C. Afterward, the solution was maintained for 5 h at 0–5 °C to complete the reaction. Lastly, water was removed by a rotary evaporator to give a colorless liquid.

### 3.4. Fabrication of Poly(p-Phenylenediamine)@Fe_3_O_4_ in [HPy][HSO_4_] Ionic Liquid 

The magnetic poly(*p*-phenylenediamine)@Fe_3_O_4_ (P*p*PDA@Fe_3_O_4_) was fabricated through the in-situ chemical oxidative polymerization in presence of iron oxide nanoparticles and [HPy][HSO_4_] ionic liquid as follows:

1 g of *p*PDA monomer was dissolved in the 30 mL of distilled water under constant stirring at room temperature and then 9.24 mmol of [HPy][HSO_4_] (optimized amount) was added to the solution. In a separate beaker, iron oxide nanoparticles (0.05 g) in 15 mL of distilled water were dispersed under ultrasonic irradiation for 30 min. Then, the iron oxide nanoparticles mixture was added to the above solution. The polymerization was initiated by the addition of 10 mL of the ammonium persulfate solution (0.99 mol/L) under constant stirring at room temperature. The mixture was retained under constant stirring at room temperature for 24 h. The precipitate was collected by the external magnet and washed with deionized water and methanol and dried at 70° for 24 h (Figure 2b). For a better comparison of the catalytic activity of nanocomposite, P*p*PDAs in the presence and absence of [HPy][HSO_4_] were also synthesized according to the above procedure. 

### 3.5. Overall Route for the Synthesis of Polyhydroquinoline Derivatives

One-pot synthesis of polyhydroquinoline compounds was carried out as follows: 

A mixture of dimedone (1.0 mmol), aldehyde (1.0 mmol), benzyl acetoacetate (1.0 mmol), ammonium acetate (1.0 mmol), and P*p*PDA@Fe_3_O_4_ (0.04 g) in ethanol solvent (5 mL) was refluxed. Rection was traced by thin-layer chromatography (hexane/ethyl acetate 5:1). Once the reaction was completed the catalyst was separated easily by an external magnet. Afterward, the crude solid product was filtered and then purified by recrystallization from ethanol. 

### 3.6. Antioxidant Activity

Antioxidant activity evaluation of prepared materials was studied in ethanolic DPPH solution (25 μM/L) by a UV-vis spectroscopy. The amount of each sample (10 mg) was added to tubes containing2 mL of ethanolic DPPH and then the tubes were kept in a dark place for 6 h. After that, DPPH inhibition (%) was measured by the following Equation:DPPH inhibition (%) = (A_b_ − A_s_)/A_b_ × 100(1)

In this equation, A_b_ and A_s_ are the absorption of DPPH solution and samples at 517 nm, respectively.

### 3.7. Antibacterial Activity

Kirby–Bauer disc diffusion technique was employed for antibacterial activities study of the prepared samples. Sample solutions (20 mg in 10 mL dimethyl sulfoxide, DMSO) were filtered by a Ministart (Sartorius). The antibacterial activity of the samples was evaluated against *Bacillus subtilis* PTCC 1023 (Gram-positive) and *Escherichia coli* PTCC 1330 (Gram-negative) bacterial species. The bacteria phase was prepared via inoculating of the cultures 1% (*v*/*v*) into the Muller–Hinton broth and incubating on a shaker at 37 °C for 24 h. Sterile paper discs were soaked with 10 μL of the sample solutions then allowed to dry. The soaked discs were placed on the agar plate and incubated at 37 °C for 24 h. The antibacterial activities of the compounds were compared with gentamicin and chloramphenicol antibiotics as positive control and DMSO as a negative control. Antibacterial activity was studied by evaluating the inhibition zone diameter (mm) of the surface of the plates and the results were reported as Mean ± SD after three repeats.

### 3.8. Characterization

The chemical structure of the synthesized materials was investigated by a Fourier transform infrared spectroscopy (FTIR) (Bruker Tensor 27, Bremen, Germany), hydrogen and carbon nuclear magnetic resonance spectroscopy (^1^HNMR and ^13^CNMR) (Bruker Avance DRX-400, Bremen, Germany), elemental analysis (CHNS) (Costech-Italy) and energy dispersive X-ray (EDX) (MIRA 3-XMU, Brno, Czech Republic). The surface morphology and crystallinity of products were evaluated by field emission scanning electron microscopy (FESEM) (MIRA 3-XMU, Czech Republic) and X-ray diffraction (XRD) (BrukerD8 Advance X-ray diffractometer, Bremen, Germany), respectively. 

## 4. Conclusions

Antibacterial and antioxidant P*p*PDA@Fe_3_O_4_ nanocomposite was successfully fabricated by in-situ oxidative polymerization in the presence of [HPy][HSO_4_] and iron oxide nanoparticles as a potential heterogeneous nanocatalyst for the synthesis of polyhydroquinolines derivatives. The nanocatalyst was characterized by different techniques and results displayed that the nanocatalyst showed superparamagnetic behavior with crystalline nature. The solubility test showed that prepared P*p*PDA in presence of [HPy][HSO_4_] had better solubility than PpPDA. The P*p*PDA@Fe_3_O_4_ nanocatalyst showed good antibacterial activity against *Escherichia coli* and *Bacillus subtilis*. The FESEM of nanocatalyst showed the hexagonal structure with a high agglomerate with a diameter of ~100 nm. The P*p*PDA@Fe_3_O_4_ nanocatalyst showed great catalytic performance in the synthesis of polyhydroquinolines derivatives and the corresponding products were synthesized with high yield (90–97%) without a difficult work-up procedure. Moreover, the P*p*PDA@Fe_3_O_4_ nanocatalyst separated easily from the reaction media by a magnet. Reusability results showed that the nanocatalyst could use for at least five times without a significant decrease in catalytic activity. According to the proposed mechanistic scheme, the prepared P*p*PDA@Fe_3_O_4_ nanocatalyst in [HPy][HSO_4_] played an important role in directing the synthesis reaction of polyhydroquinolines derivatives with favorable features, e.g., Brønsted acid, strong basic sites, and high surface area. It could be concluded the bioactive P*p*PDA@Fe_3_O_4_ nanocomposite could be employed as an eco-friendly and high efficiency nanocatalyst for the synthesis of different organic reactions. P*p*PDA@Fe_3_O_4_ nanocatalysts and 11 polyhydroquinolines derivatives showed antioxidant activity between 75% and 99%. Among polyhydroquinolines derivatives, only 5r and 5v had good growth inhibitory effects against tested microorganisms.

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
