# Peer review of "Ionic Liquid-Assisted Fabrication of Bioactive Heterogeneous Magnetic Nanocatalyst with Antioxidant and Antibacterial Activities for the Synthesis of Polyhydroquinoline Derivatives"

_molecules, 2022, doi:10.3390/molecules27051748_

Round 1

Reviewer 1 Report

The presented study introduces an interesting research about novel iron oxide-based nanocomposite with antibacterial effect against Gram + and Gram – bacteria. The application of ionic liquids for the preparation of magnetic nanostructures can mean a green alternative for nanobiotechnological process.

The structure of the experimental work is clean, the manuscript is well-built the introduction provides enough information related to the study.

Only few comments can be added to this work, which are the following:

  • Scheme is essential for this paper, however some information should be added, while some part of the figure is not clear enough. Author should label the nanocatalysts and show the role of it in two application (antioxidant and antibacterial routes).

  • In my opinion, Scheme 2 is also important to highlight at the beginning of the paper. Perhaps the merging of Scheme 1 and 2 could be a good solution for better understanding of the main messages of the study.

  • In Fig2, a microscopic image about individual magnetic particles should be also useful.

  • How many parallel measurements were performed with SEM-EDAX? Are there any information about the standard error of the results presented in Table 1 ? Why was the Fe content not determined? The difference between results in Table 1 and Fig1b is not clearly seen.

  • Is the measuring of “inhibition zones in mm” for the determination of antibacterial activity adequate method? Have the authors information about CFU numbers? Or did you do other method for the determination of the ration of living and dead bacterial cells? How many parallel experiments were done in spreading experiments?

After answer the questions and revising the manuscript according to above mention comments, it could be accepted for publication in Molecules.

Author Response

The presented study introduces an interesting research about novel iron oxide-based nanocomposite with antibacterial effect against Gram + and Gram – bacteria. The application of ionic liquids for the preparation of magnetic nanostructures can mean a green alternative for nanobiotechnological process.The structure of the experimental work is clean, the manuscript is well-built the introduction provides enough information related to the study. Only few comments can be added to this work, which are the following:

RESPONSE: The authors greatly appreciate your consideration of the manuscript and thank you for the encouraging remarks and your valuable comments to improve the quality of the paper.

  1. Scheme is essential for this paper, however some information should be added, while some part of the figure is not clear enough. Author should label the nanocatalysts and show the role of it in two application (antioxidant and antibacterial routes).

Response: Thank you for your comment. Scheme 1 was revised.

  1. In my opinion, Scheme 2 is also important to highlight at the beginning of the paper. Perhaps the merging of Scheme 1 and 2 could be a good solution for better understanding of the main messages of the study.

Response: Thank you for your comment. Scheme 1 was revised and some items of Scheme 2 were added to Scheme 1.

  1. In Fig2, a microscopic image about individual magnetic particles should be also useful.

Response: The FESEM of Fe3O4 nanoparticles was added to Fig.2

  1. How many parallel measurements were performed with SEM-EDAX? Are there any information about the standard error of the results presented in Table 1 ? Why was the Fe content not determined? The difference between results in Table 1 and Fig1b is not clearly seen.

Response: Thank you very much for your valuable comment. In addition to FTIR and XRD analyses, we also used the elemental and EDX analyses to prove the preparation of the polymer and nanocomposite. As you know, the results of EDX are different from the results of elemental analysis because EDX analysis is a more qualitative analysis but the results of elemental analysis are more accurate. In addition, the “expected %", amount of elements were added to Table 1.

  1. Is the measuring of “inhibition zones in mm” for the determination of antibacterial activity adequate method? Have the authors information about CFU numbers? Or did you do other method for the determination of the ration of living and dead bacterial cells? How many parallel experiments were done in spreading experiments?

Response: Thank you for your valuable comment. To determine the inhibitory effect of synthesized prepared materials on pathogenic bacteria can be the use of agar well diffusion test and microdilution test (to determine minimum inhibitory concentration-MIC) as primary and confirmatory tests, respectively. Agar media (tryptic soy agar and Muller Hinton agar) and broth media (tryptic soy broth and Muller Hinton broth) were used for them, respectively.  In principle, the agar well diffusion test was used as the primary test to determine which compounds possess antibacterial activity. In the current study, we use of agar disc diffusion test and results are repeated three times and the standard deviation of obtained results is also calculated (Table 6)

Reviewer 2 Report

Manuscript ID: molecules-1633090

Title: Ionic Liquid-assisted Fabrication of Bioactive Heterogeneous Magnetic
Nanocatalyst with Antioxidant and Antibacterial Activity for the Synthesis of
Polyhydroquinoline Derivatives

Reviewer’s comments:

The paper describes in detail the synthesis of a new nanocomposite,

 poly(p-phenylenediamine)@Fe3O4, by the in-situ polymerization in the presence of [HPy][HSO4] and iron oxide nanoparticles. The obtained nanocomposite was successfully tested for antibacterial properties against Gram-negative (Escherichia coli) and Gram-positive (Bacillus subtilis) bacteria. The antioxidant activity of Fe3O4 NPs, PpPDA,  PpPDA@[HPy][HSO4], PpPDA@Fe3O4@[HPy][HSO4], and synthesized polyhydroquino derivatives was studied, with positive results, concluding that they may play a positive role in the synthesis of immune-boosting drugs.

Observations:

  1. DDPH is firstly mentioned on page 13, raw 255, but  DPPH  (2,2-diphenyl-1-picrylhydrazyl) is explicit on raw 416, page 18. It should be explained when firstly appearing in the text, that is on page 13.
  2. raw 255: Correct form: “was studied”, because the subject is “the antioxidant activity”.
  3. In Introduction, in the revised manuscript, to enrich the domain of the research background, the authors are invited to read and add as reference the paper:

Donescu, D., Fierascu, R.C., Ghiurea, M., Manaila-Maximean, D., Nicolae, C.A., Somoghi, R., Spataru, C.I., Stanica, N., Raditoiu, V. and Vasile, E., 2017. Synthesis and magnetic properties of inverted core-shell polyaniline-ferrite composite. Applied Surface Science, 414, pp.8-17. DOI: 10.1016/j.apsusc.2017.04.061

In Conclusion, I recommend publication after minor corrections.

Author Response

The paper describes in detail the synthesis of a new nanocomposite, poly(p-phenylenediamine)@Fe3O4, by the in-situ polymerization in the presence of [HPy][HSO4] and iron oxide nanoparticles. The obtained nanocomposite was successfully tested for antibacterial properties against Gram-negative (Escherichia coli) and Gram-positive (Bacillus subtilis) bacteria. The antioxidant activity of Fe3O4 NPs, PpPDA,  PpPDA@[HPy][HSO4], PpPDA@Fe3O4@[HPy][HSO4], and synthesized polyhydroquino derivatives was studied, with positive results, concluding that they may play a positive role in the synthesis of immune-boosting drugs.

RESPONSE: The authors greatly appreciate your consideration of the manuscript and thank you for the encouraging remarks and your valuable comments to improve the quality of the paper.

Observations:

  1. DDPH is firstly mentioned on page 13, raw 255, but  DPPH  (2,2-diphenyl-1-picrylhydrazyl) is explicit on raw 416, page 18. It should be explained when firstly appearing in the text, that is on page 13.

Response: The full name of DPPH was added to Page 13.

  1. raw 255: Correct form: “was studied”, because the subject is “the antioxidant activity”.

Response: It was corrected.

  1. In Introduction, in the revised manuscript, to enrich the domain of the research background, the authors are invited to read and add as reference the paper:

Donescu, D., Fierascu, R.C., Ghiurea, M., Manaila-Maximean, D., Nicolae, C.A., Somoghi, R., Spataru, C.I., Stanica, N., Raditoiu, V. and Vasile, E., 2017. Synthesis and magnetic properties of inverted core-shell polyaniline-ferrite composite. Applied Surface Science414, pp.8-17. DOI: 10.1016/j.apsusc.2017.04.061

Response: The mentioned reference was added to the introduction section.

Reviewer 3 Report

The paper reports on the synthesis of a family of polyhydroquinolines with antibacterial properties. The synthesis is based on heterogeneous catalysis by a magnetic nanocatalyst. A number of compounds were prepared, characterized and tested for antibacterial activity. 
The amount of work is noteworthy. However, major revisions are necessary before publication.

In Introduction the authors state that “ materials with antioxidant and antibacterial activities can cause toxicity due to their low biocompatibility and safety profile, urging scientists to follow new ways in synthesis of such materials.” However there is no mention of biocompatibility and safety profile of the compounds used here.

Line 228 and Figure 3 speak of “proposed mechanism”. A mechanism must follow adequate experimental verification. This is only “paper chemistry”. Reasonable, but not based on ad hoc experiments. For example, it is hard to believe the presence of free ammonia in the reaction conditions. I suggest to use the term “mechanistic scheme” instead of “mechanism”: it is more correct.  

Language must be accurately checked. Also revision of presentation is necessary. Selected (not exhaustive) examples are in the following

Line 24 “… The chemical preparation of PpPDA@Fe3O4  nanocomposite was initiated through the spontaneous oxidation of p-phenylenediamine by ammonium persulfate …” I am not sure that “spontaneous” is the correct word, considering the intervention of ammonium persulfate.

What do you mean with "cargo delivery purposes" (line 66)?

line 112 "for confirming the preparation of the prepared..." sounds funny 

Table 1.  Generally, it is good habit to list not only "found %", but also "expected %", otherwise found values are meaningless or it is impolite to force the reader to calculate the expected values.

line 180 and Table 3 n-hexane. The prefix n- for linear chains has been abolished long time ago by IUPAC because it is unnecessary. I know it is still (mistakenly) used, but let’s begin to follow the rule and avoid it!

Table 3 the heading Yield has (%b), but there is no b as footnote. Instead, it is present in Table 4, but the closing parenthesis is superscript (%b).

Table 5 I do not see the need of the column "reported" for the melting points, since none of them is reported. A footnote is enough. By the way, the superscript c does not appear in the footnote. Please, note that "NR" is repeatedly written in the column Ref. (instead of "current study"). Also this column is superfluous, since there are not different references.

Why lines from 287 to 292 are written in italic?

line 430 "Lastly, water was removed by a rotary evaporator to give a colorless liquid" In my year-long experience with Ionic Liquids this sentence is hard to believe! You were really lucky! Generally a yellow liquid is the result. 

line 496  "in compared" better "than". 

Author Response

The paper reports on the synthesis of a family of polyhydroquinolines with antibacterial properties. The synthesis is based on heterogeneous catalysis by a magnetic nanocatalyst. A number of compounds were prepared, characterized and tested for antibacterial activity. The amount of work is noteworthy. However, major revisions are necessary before publication.

RESPONSE: The authors greatly appreciate your consideration of the manuscript and thank you for the encouraging remarks and your valuable comments to improve the quality of the paper.

In Introduction the authors state that “ materials with antioxidant and antibacterial activities can cause toxicity due to their low biocompatibility and safety profile, urging scientists to follow new ways in the synthesis of such materials.” However there is no mention of the biocompatibility and safety profile of the compounds used here.

Response: Response: In this work, due to the high volume of data, data on cytotoxicity of samples were not reported and only their use as effective nanocatalysts for the manufacture of drug derivatives was investigated. Our results (not shown here) displayed that the nanocatalyst used in this project has very low cytotoxicity and the survival rate of cells after 72 hours is over 85%. The study of the biological properties of the synthesized derivatives such as anti-cancer test and their cytotoxicity will be reported in the future work of our group. We have studied and compared the biological properties of poly (phenylenediamine) derivatives such as cell toxicity, anticancer, and photothermal activity of them have been submitted to another journal. The aim of this work is the use of bioactive magnetic poly (p-phenylenediamine)/IL as heterogeneous catalysts in the synthesis of poly-hydro quinoline derivatives.

Line 228 and Figure 3 speak of “proposed mechanism”. A mechanism must follow adequate experimental verification. This is only “paper chemistry”. Reasonable, but not based on ad hoc experiments. For example, it is hard to believe the presence of free ammonia in the reaction conditions. I suggest to use the term “mechanistic scheme” instead of “mechanism”: it is more correct.  

Response:  Thank you for your valuable comment. It was corrected based on your comment.

Language must be accurately checked. Also revision of presentation is necessary. Selected (not exhaustive) examples are in the following

Response:  The language of the manuscript was rechecked carefully.

Line 24 “… The chemical preparation of PpPDA@Fe3O4  nanocomposite was initiated through the spontaneous oxidation of p-phenylenediamine by ammonium persulfate …” I am not sure that “spontaneous” is the correct word, considering the intervention of ammonium persulfate.

Response:  Thank you for your valuable comment. It was corrected based on your comment.

What do you mean with "cargo delivery purposes" (line 66)?

Response: We meant that magnetic nanoparticles could be used as carrier for drugs. Superparamagnetic iron oxide nanoparticles (SPIONs) are the most extensively studied inorganic nanocarrier systems for drug delivery [1].

Ref.

[1] Marika Musielak Igor Piotrowski, Wiktoria M.Suchorska, Superparamagnetic iron oxide nanoparticles (SPIONs) as a multifunctional tool in various cancer therapies, Reports of Practical Oncology & Radiotherapy Volume 24, Issue 4, July–August 2019, Pages 307-314

line 112 "for confirming the preparation of the prepared..." sounds funny 

Response: It was corrected.

Table 1.  Generally, it is good habit to list not only "found %", but also "expected %", otherwise found values are meaningless or it is impolite to force the reader to calculate the expected values.

Response: Thank you for your comment. The “expected %” data were added to Table 1.

line 180 and Table 3 n-hexane. The prefix n- for linear chains has been abolished long time ago by IUPAC because it is unnecessary. I know it is still (mistakenly) used, but let’s begin to follow the rule and avoid it!

Response: Thank you for your comment. It was corrected

Table 3 the heading Yield has (%b), but there is no b as footnote. Instead, it is present in Table 4, but the closing parenthesis is superscript (%b).

Response: Thank you for your comment. It was added to Table 3

Table 5 I do not see the need of the column "reported" for the melting points, since none of them is reported. A footnote is enough. By the way, the superscript c does not appear in the footnote. Please, note that "NR" is repeatedly written in the column Ref. (instead of "current study"). Also this column is superfluous, since there are not different references.

Response: The mentioned columns were removed.

Why lines from 287 to 292 are written in italic?

Response: It was corrected.

line 430 "Lastly, water was removed by a rotary evaporator to give a colorless liquid" In my year-long experience with Ionic Liquids this sentence is hard to believe! You were really lucky! Generally a yellow liquid is the result. 

Response: We prepared [HPy][HSO4] IL according to the following reference.

Tao, D. J.; Wu, Y. T.; Zhou, Z.; Geng, J.; Hu, X. B.; Zhang, Z. B. Kinetics for the esterification reaction of n -butanol with acetic acid catalyzed by noncorrosive brønsted acidic ionic liquids. Ind. Eng. Chem. Res. 2011, 50, 1989–1996, doi:10.1021/ie102093e.

line 496  "in compared" better "than".

Response: It was corrected.